# The Impact of the COVID-19 Pandemic on the Revealed Comparative Advantage of Industries in the Baltic States

Jolanta Droždz *, Arūnas Burinskas and Viktorija Cohen

Faculty of Economics and Business Administration, Vilnius University, 10222 Vilnius, Lithuania
* Correspondence: jolanta.drozdz@cr.vu.lt

**Abstract:** The active spread of COVID-19 and the resulting containment measures have made it clear that both supply and demand and global production networks are facing unprecedented shocks and disruptions. Accordingly, this has resulted in an urgent need to investigate countries' competitive situation (and its changes) during a prolonged period of uncertainty. This study aims to assess the impact of the COVID-19 pandemic on the revealed comparative advantage (RCA/ Balassa index) of the Baltic states' industries. The Balassa index was calculated for nine industries in three Baltic States (Lithuania, Latvia, and Estonia). The RCA index calculations were linked to COVID-19 cases in the countries under consideration by forming a regression equation. RCA index values and their changes were evaluated by analysing data before and after the pandemic, covering the period between 2017 and 2021. This study revealed that the COVID-19 pandemic's impact on each Baltic country's competitiveness in trade with EU countries is significantly higher than in trade with third countries. The results show that Baltic states did not have a comparative advantage in trade with third countries during this time. However, Lithuania and Latvia proved to be more resilient to the consequences of the pandemic, even though industries with a low RCA were more affected. Meanwhile, in trade with EU countries, many of the Baltic states' industries appeared to have a comparative advantage, which began to decline a few years before the pandemic's start. Nevertheless, highly competitive Baltics industries showed remarkable resilience to the impact of the pandemic. However, a short-term decrease in the RCA was observed in individual cases.

**Keywords:** COVID-19 pandemic; international trade; revealed comparative advantage; RCA index; Balassa index; competitiveness; Baltic states

## 1. Introduction

The currently-used term "pandemic" is a new concept, the definition of which has not been established in the legislation of many countries. From a scientific point of view, the impact of different pandemics (before COVID-19) was not a frequently analysed topic, and various coronavirus outbreaks (e.g., MERS-CoV or SARS-CoV) were controlled quickly enough at the national or regional level. However, some studies have previously argued that the possibility of a global pandemic might be a growing concern (Garett 2008). This can be attributed to the rapid development of globalisation for several decades, the openness of economies contributing to strengthening ties between individual countries or regions, and the growth of the global population. Any estimate of the impact of a possible pandemic is associated with significant uncertainties, and the interpretation and modelling of existing data cannot predict the probability of a pandemic (Meltzer et al. 1999). Meanwhile, relatively little academic attention has been paid to the serious evaluation of a pandemic's economic consequences—possibly due to data limitations (Garett 2008). The COVID-19 pandemic forced a new look at the relations and dependencies between countries, disrupted the regular order, demanded quick solutions, and continues to require a deeper analysis and more accurate knowledge of the impact caused by the unexpected and unprecedented crisis. It can already be stated that the COVID-19 pandemic will have tremendous consequences

for countries' economies and international trade. However, as these effects have not yet been studied deeply enough, more detailed studies are necessary to help governments formulate retaliatory economic policy measures.

It is already clear that the negative economic consequences of the COVID-19 pandemic will differ between counties and sectors, both in terms of production, international trade, and the attraction of foreign direct investment (FDI). Therefore, sectoral studies of the manufacturing industry and international trade can be of a particular importance in the formation of both general economic policies and specific government industrial measures within the legal framework of the EU's business support policy.

The COVID-19 virus reached and spread through the European continent in the first six months of 2020 (Bolt et al. 2021), causing considerable confusion and great uncertainty. The active spread of COVID-19 and the resulting containment measures resulted in supply and demand and global production networks experiencing shocks and disruptions at a hitherto unprecedented scale. Since the beginning of the pandemic, various scientific studies have predicted that, as the crisis continues, adverse changes in consumer expectations, industrial production, and retail income (Fernandes 2020) will have long-term negative consequences (IMF 2020; World Bank 2020a). The impact of the COVID-19 pandemic on central and eastern European countries (CEECs) was delayed and weaker than in older EU member states due to the CEECs' specialisation in advanced manufacturing and services (Nacewska-Twardowska 2021). As a result, they are more dependent on imported inputs for exports (World Bank 2020a; 2020b). Therefore, more comprehensive studies of international trade flows are relevant and essential. The competitive position of CEECs in world markets depends on smooth participation in international trade, access to global resources, and active participation in international value chains.

As a part of the EU, Baltic countries have their specifics, which have not been sufficiently reflected in studies covering CEECs (Brada et al. 2021; Hajduova et al. 2021; Nacewska-Twardowska 2021; Dauti and Elezi 2022; Kuzmenko et al. 2022) or the EU in general (Boikova et al. 2021). Only a handful of studies, e.g., (Pilinkienė 2014, 2015; Zabotkina et al. 2020; Bolt et al. 2021; Dzemydaitė 2021; Petrylė 2022a, 2022b), have concentrated on the specifics of the Baltic states and their competitiveness to identify the impact of the COVID-19 pandemic and determine its consequences for the manufacturing industry. However, more detailed empirical studies on the pandemic's impact on the economies and international trade of the Baltic countries are lacking, and this region has been underrepresented in the scientific literature. In sum, it can already be stated that the COVID-19 pandemic will have unprecedented consequences for countries' economies and international trade; they have not yet been afforded sufficient academic attention. Therefore, detailed studies are necessary to help governments form necessary retaliatory economic measures.

In this context, the impact of the COVID-19 pandemic on the Baltic countries' resilience to shocks and crises has been understudied. Indeed, there is scant empirical evidence on the different economic sectors and their situation in the context of the COVID-19 pandemic. As in many countries, a direct impact has been observed on accommodation and catering establishments (thus affecting the tourism sector). Still, there is a lack of information on the situation in other sectors of the economy—particularly in the Baltic states. Therefore, this study aims to assess the impact of the COVID-19 pandemic on the revealed comparative advantage (RCA) of the industries in the Baltic states. Scholars clearly understand that improving national competitiveness is fundamental to raising long-term economic growth and enhancing living standards (Boikova et al. 2021). Therefore, this research seeks to assess the competitive position of the manufacturing industry sectors of the Baltic countries in the development of international trade both before and after the start of the COVID-19 pandemic. In so doing, this study intends to identify the signals of change and explain their causes.

The topic under consideration is relevant for both practical and theoretical reasons. Indeed, potential recommendations for economic policies and scientific contributions to theories of international trade can be revealed through its study. First, governments should

address the consequences of international and domestic political uncertainty by assessing its causes, thereby helping to shape the resilience of the country's economy to external shocks. Resilience is understood as the ability to resist, absorb, recover from, or successfully adapt to adversity or a change in conditions (OECD 2014). In terms of recommendations for governments, the first to mention is the extraordinary challenges that require a crisis management approach, that is, a quick response of both the government and businesses to market processes in combating the economic consequences caused by the COVID-19 pandemic.

## 2. Literature Review

Since the outbreak of the pandemic, many studies have been conducted to assess its likely impact, with a particular emphasis on modelling the global situation (Freeman and Baldwin 2020; Baldwin and Mauro 2020; Maliszewska et al. 2020; Vidya and Prabheesh 2020; Arriola et al. 2021; Hayakawa and Mukunoki 2021; Vo and Tran 2021; Xing et al. 2021; Nacewska-Twardowska 2022) and conducting more detailed assessments of individual regions (Demirguc-Kunt et al. 2020; Zabotkina et al. 2020; Boikova et al. 2021; Bolt et al. 2021; Hajduova et al. 2021; Nacewska-Twardowska 2021; Dauti and Elezi 2022). In the scope of individual studies, some sectors have been afforded more attention due to their importance to the economy (Maliszewska et al. 2020; Ayat et al. 2021; Iqbal et al. 2021; Gajdosikova et al. 2022). Country-specific cases have been explored extensively (Duan et al. 2020; Minondo 2021; Barichello 2021; Černikovaitė and Karazijienė 2021; Davidescu et al. 2021; Du and Shepotylo 2021; Svabova et al. 2021; Ullah et al. 2021; Wosiek and Visvizi 2021; Zimon and Dankiewicz 2020; Costa et al. 2022; Gajdosikova et al. 2022; Khan Jaffur et al. 2022; Kuzmenko et al. 2022; Nacewska-Twardowska 2022; Petrylė 2022a, 2022b; Purwono et al. 2022). Within the existing literature, more specific aspects of research related to the COVID-19 pandemic can also be discovered, such as consumer behaviour (Borsellino et al. 2020; Šneiderienė et al. 2020; Černikovaitė and Karazijienė 2021; Valaskova et al. 2021), government communication strategies during the pandemic (Bolt et al. 2021), or the improvement of trading strategies (Guobužaitė and Teresienė 2021); on challenges in ensuring global access to COVID-19 vaccines (Wouters et al. 2021), and the role of logistics in times of uncertainty (Li et al. 2023).

Previously, scholars argued that any pandemic's most significant adverse economic impact would be associated with mortality (Brahmbhatt 2005). However, the effect of the global COVID-19 pandemic has manifested itself in various ways, not only in terms of claiming human lives. Macroeconomic factors, such as employment and the labour market, GDP dynamics, state debt analysis of partner markets, the evaluation of FDI flows (Brada et al. 2021; Davidescu et al. 2021; Svabova et al. 2021), and the impact of fiscal deficit on economic growth (Bolt et al. 2021; Dauti and Elezi 2022; Maliszewska et al. 2020; Ajmal et al. 2021) have become significant in research. Coquidé et al.'s (2022) study demonstrated a strong impact of COVID-19 on international trade. The authors argued that there are multiple social and economic origins of this negative impact. In addition, some countries have more effectively managed the pandemic and preserved individual economic sectors than others. Indeed, different responses to the COVID-19 pandemic have been observed among various countries, such as the immediate reaction of CEECs (Nacewska-Twardowska 2021).

During the first stages of the pandemic, the initial reaction and research of economists were focused on assessing the possible macroeconomic condition, the forecasting of which was hindered by the level of uncertainty and the nonexistence of specific models (which have since been developed). At one point, while many countries around the world experienced similar limitations and restrictions, the associated consequences were somewhat different. However, scholars agreed that, with wide-scale limits on the movement of people and goods, COVID-19 effects were expected in several directions, such as direct supply disruptions, supply chain disruptions, and demand disruptions due to macroeconomic drops and wait-and-see purchase delays (Baldwin and Mauro 2020; Baqaee and Farhi

2021). This contributed to a global slowdown (Freeman and Baldwin 2020; Xing et al. 2021; Nacewska-Twardowska 2021).

Regarding the supply side, the pandemic caused a reduction in the scale of production and the export supply in particular countries due to uncertainties. COVID-19 restrictions reduced firms' ability to maintain production at pre-pandemic prices and quantities (Baqaee and Farhi 2021). Exports were expected to drop, particularly in industries and countries where remote work/operation was less feasible (Hayakawa and Mukunoki 2021). COVID-induced restrictive measures were, or still are, aimed at reducing or completely halting the process of shops and services or, in some cases, large manufacturing companies (Svabova et al. 2021). Services and tourism were the hardest hit sectors, whereas manufacturing and agriculture faced less decline (Maliszewska et al. 2020). Labour-intensive industries were observed to be more likely to suffer from the adverse effects of COVID-19 (Maliszewska et al. 2020; Hayakawa and Mukunoki 2021).

The impact of COVID-19's damage in importing countries appeared mainly due to the decrease in aggregate demand (Hayakawa and Mukunoki 2021). These negative shocks affected different industries differently: whereas some producers easily switched to remote work and maintained both employment and production, industries that required face-to-face contact were forced to reduce production capacity and employment (Baqaee and Farhi 2021). The demand side has been disturbed by heightened unemployment (Ullah et al. 2021), which caused reductions in people's earnings and purchasing power, minimised their visits to retail outlets (Hayakawa and Mukunoki 2021), and changed shopping habits and future expectations (Gajdosikova et al. 2022), all of which resulted in reduced spending (Baqaee and Farhi 2021). COVID-19 has affected the economic sphere and health, healthcare, safety, and health protection in the workplace (Gajdosikova et al. 2022). Moreover, it has increased poverty and even raised illiteracy levels in developing countries (Ullah et al. 2021).

Regional and global value chains, as well as regional development (Ullah et al. 2021; Nacewska-Twardowska 2021, 2022; Purwono et al. 2022), global value chains, and their success factors (Cieślik et al. 2016, 2019; Duan et al. 2020; Nikulin and Szymczak 2020; Baqaee and Farhi 2021; Bolt et al. 2021; Nacewska-Twardowska 2021; Espitia et al. 2022; Li et al. 2023) have become some of the most important topics discussed in the academic community in recent years. However, it is worth mentioning that the speed with which supply chain actors have been able to reorganise themselves to ensure food availability, at least in the developed world (OECD 2020), has been truly remarkable. An OECD report (2020) stated that, generally speaking, food supply chains in the developed world have demonstrated outstanding robustness and resilience in the face of COVID-19.

Different authors have emphasized regional trade as an essential factor of strong trade ties and the success factor in reacting to unforeseen circumstances (Brada et al. 2021; Nacewska-Twardowska 2021; Ullah et al. 2021; Purwono et al. 2022). It should be further mentioned that global networks were under pressure even before the COVID-19 pandemic (Xing et al. 2021). The world economy felt a decline in global demand, which had begun to manifest itself, especially toward the end of 2019 (Gajdosikova et al. 2022). According to economists (e.g., Susskind et al. 2020), the world economy is expected to change after the COVID-19 crisis in the following ways. For example, global supply chains will be more regional, data flows across national borders will grow explosively, and remote work will become even more popular.

The consequences of the COVID-19 pandemic are interwoven with the digitalisation of business, the Industry 4.0 revolution, the relocation of global manufacturing from China, the global redistribution of supply chains, new agreements between economic unions, and other processes that promote or limit globalisation. It is noteworthy that the COVID-19 pandemic has brought opportunities and challenges (Duan et al. 2020). Indeed, Chamola et al. (2020) argued that the pandemic has already had a significant positive impact on the development of such information technologies (IT) as 5G. It is believed that this will inevitably accelerate the Industry 4.0's revolution, which is centred on 5G, the Internet

of Things, and cyber–physical systems (CPS) (Lu 2017). Additionally, it can be expected that trends in the redistribution of economic power globally will accelerate (World Bank Group 2015), which will inevitably lead to changes in international trade. For example, rising wage inflation in China's manufacturing sector may shift the "global manufacturing factory" to countries closer to export markets (PWC 2020), and remote work options will bring the world even closer together and become a common practice (Susskind et al. 2020).

The RCA index (Balassa 1965) is a classical and widely used measure of a country's competitiveness and export position (Costinot 2009; Laursen 2015; Sposi 2015; Bernatonytė et al. 2013; Serva and Vitunskienė 2014; Pilinkienė 2014; Laursen 2015; Drożdz 2018; Wosiek and Visvizi 2021; Kuzmenko et al. 2022; Purwono et al. 2022). Additional empirical studies on factors driving changes or explanations in RCA and specialization patterns were covered by Balassa and Noland's (1988) and Hoang et al.'s (2017) studies. It indicates performance concerning other markets and illustrates a country's internal implementation. While competitiveness does not have a unified definition (Olczyk 2016; Boikova et al. 2021) and can be analysed on different levels (e.g., regional, sectoral, industry, or company level), we opted to follow Krugman's (1994) national competitiveness theory and evaluate competitiveness on the macro level, which is associated with improved living standards and determined by raising productivity that creates the competitive advantage of a country in international trade. According to the OECD (2005), competitiveness should be understood as an economy's ability to compete fairly and successfully in global markets for goods and services, improving the living standards of a given country's citizens. Moreover, the OECD (2001) has defined competitiveness in international trade as a measure of a country's advantage or disadvantage in selling its products in global markets.

Export best works in tandem with productivity, which internally encourages companies to restructure their resources efficiently for higher international trade competitiveness (Gaglio 2015; Nowak et al. 2020). Boikova et al. (2021) also emphasised the importance of competitiveness performance and its determinants, which change along with macroeconomic factors, business environment, and consumer demand. However, nearly most competencies, resources, technologies, market opportunities, and information can be replicated by competitors, and thus no competitive advantage can remain permanent. On the other hand, countries and companies can gain and duplicate specific skills and improve their market competitiveness (Ilinova et al. 2021). In addition, the abovementioned RCA index, which is most often used as an indicator of specialisation in international trade, has been adapted to sector analysis (Bernatonytė et al. 2013; Serva and Vitunskienė 2014; Pilinkienė 2014; Laursen 2015; Drożdz 2018; Wosiek and Visvizi 2021; Kuzmenko et al. 2022).

### 3. Materials and Methods

This study seeks to assess the changes in the competitive advantage of the Baltic states under consideration (Estonia, Latvia, and Lithuania) that have arisen under the influence of the COVID-19 pandemic. As has been widely established, the most effective rating of competitive advantage is in terms of autarky relative cost, which, in practice, is impossible to observe since countries have long been engaged in international trade. For this reason, we applied the RCA methodology proposed by Balassa (1965), according to which the share of a country's exports in a particular sector is compared with the total percentage of its exports in the total volume of international trade:

$$RCA_{ij} = 100 \frac{X_{ij}/X_{wj}}{X_{it}/X_{wt}} \tag{1}$$

where $X_{ij}$ and $X_{wj}$ are the export of good *j to* the country *i* and the world *w*; and $X_{it}$, $X_{wt}$ are their total export.

It is essential to note that *RCA* > 100 and higher serves as an indicator of increasing competitive advantage when *RCA* < 100 shows that the country does not have a competitive advantage and is declining in the specific industry. RCA index is used to determine country specialisation as a structural indicator of trade flows. Moreover, an issuer's comparative

advantage is often used to identify a country's weak and solid industries/sectors. The RCA index is calculated based on structural indicators of international trade. A country's relatively more significant share in export markets reveals its comparative advantage and vice versa (Andrews and Serres 2016).

We used EUROSTAT data covering Baltic countries (Estonia, Latvia, and Lithuania) for the current study's analysis. In addition, the monthly Extra-EU27 and Intra-EU27 export flow data cover January 2017 to December 2021. The data nomenclature was NO-074754 BEC/rev.5 (see Table 1).

**Table 1.** The data nomenclature we use: NO-074754 BEC/rev.5.

| No | Industry Name |
|----|---------------|
| 1 | Agriculture, forestry, fishing, food, beverages, tobacco |
| 2 | Mining, quarrying, refinery, fuels, chemicals, electricity, water, waste treatment |
| 3 | Construction, wood, glass, stone, basic metals, housing, electrical appliances, furniture |
| 4 | Textile, apparel, shoes |
| 5 | Transport equipment and services, travel, postal services |
| 6 | ICT, media, computers, business and financial services |
| 7 | Health, pharmaceuticals, education, culture, sport |
| 8 | Government, Military |
| 9 | Other goods |

Source: Eurostat.

The descriptive statistics of the Balassa RCA indexes data we used are provided in Table 2.

**Table 2.** Descriptive statistics of the RCA index of the Baltic States.

| Country | Export Direction | Mean | Median | Max. | Min. | Std. Dev | Skewness | Kurtosis | Obs. |
|---------|------------------|------|--------|------|------|----------|----------|----------|------|
| Estonia | Extra-EU27 | 15.36 | 8.03 | 242.65 | 0.00 | 19.98 | 3.58 | 33.85 | 540 |
| Estonia | Intra-EU27 | 118.40 | 117.91 | 391.76 | 3.85 | 83.16 | 0.46 | 2.90 | 540 |
| Latvia | Extra-EU27 | 19.14 | 9.08 | 140.49 | 0.00 | 24.18 | 1.80 | 6.61 | 540 |
| Latvia | Intra-EU27 | 81.10 | 77.99 | 801.63 | 1.27 | 77.06 | 2.66 | 19.68 | 540 |
| Lithuania | Extra-EU27 | 16.26 | 11.87 | 239.03 | 0.00 | 22.94 | 4.17 | 31.48 | 540 |
| Lithuania | Intra-EU27 | 54.90 | 57.81 | 207.03 | 0.05 | 40.63 | 0.51 | 3.10 | 540 |

Source: calculated by the authors.

With this, we sought to determine whether there has been a difference in the impact of COVID-19 on the Baltic countries in terms of comparative advantage. Furthermore, we tested the RCA index dynamics separately regarding trade with other EU members and third countries.

In the analysis, we employed the mean equality test of the RCA index averages to compare data from 2017 with each subsequent year up to 2021. This method tests whether subgroups have the same mean, which should be the case when the variability between the sample means (between groups) should be the same as the variability within any subgroup. In this simple model, we denoted i-th observation in subgroup $g$ as $x_{(g,i)}$, where $i = 1, \ldots, n_g$ for groups $g = 1, 2, \ldots G$. The between and within sums of squares are defined as:

$$SS_B = \sum_{g=1}^{G} n_g \left( \overline{x}_g - \overline{x} \right)^2 \tag{2}$$

$$SS_W = \sum_{g=1}^{G} \sum_{i=1}^{n_g} \left( x_{ig} - \overline{x}_g \right)^2 \tag{3}$$

where $\overline{x}_g$ is the sample mean within-group $g$ and $\overline{x}$ is the overall sample mean. The $F$-statistic for the equality of group means is computed as:

$$F = \frac{SS_B/(G-1)}{SS_W/(N-G)} \tag{4}$$

where $N$ is the number of the total observations, having an F-distribution with $G-1$ degrees of freedom of numerator, and $N$-$G$ degrees of freedom for denominator, we tested the null hypothesis of the data being independent and identically normally distributed (i.e., being equal by means and variance in each subgroup).

We used the test statistics that deal with heterogeneity in the variance of subgroups suggested by Welch (1951) with the idea of forming F-statistics in such a way as to be able to cope with the unequal variances. Using weight function $w_g = n_g/s_g^2$ where $s_g^2$ is sample variance in subgroup $g$, we were able to obtain the modified F-statistic:

$$F^* = \frac{\sum_{g=1}^{G} w_g \left(\overline{x}_g - \overline{x}^*\right)^2/(G-1)}{1 + \frac{2(G-2)}{(G^2-1)} \sum_{g=1}^{G} \frac{\left(1-h_g\right)^2}{n_g-1}} \tag{5}$$

where $h_g$—a normalised weight, $\overline{x}^*$—the weighted grand mean $h_g = w_g / \sum_{g=1}^{G} w_k$, and $\overline{x}^* = \sum_{g=1}^{G} h_k \overline{x}_g$.

At the same time, we here report the t-statistic—the square root of the $F$-statistic with one numerator degree of freedom. With this simple method, we could track statistically significant differences in yearly averages that may have occurred before and during the COVID-19 pandemic.

After identifying statistically significant changes in RCA averages, we later investigated their possible association with COVID-19 cases. So, we investigated whether COVID-19 cases could have affected competitiveness in the Baltics. Therefore, we employed an OLS regression analysis to check how COVID-19 cases in each Baltic country affected RCA in different industries. Finally, we expected to compare these results to RCA index dynamics in various industries to distinguish possible COVID-19 effects from tendencies that could persist before its outbreak. For this purpose, we used official Baltic COVID-19 case data, the descriptive statistics of which are presented in Table 3.

**Table 3.** Descriptive statistics of smoothed COVID-19 monthly cases.

| Country | Mean | Median | Max. | Min. | Std. Dev | Skewness | Kurtosis | Jarque-Bera | Prob. | Obs. |
|---|---|---|---|---|---|---|---|---|---|---|
| Estonia | 280.10 | 127.84 | 1282.67 | 0.22 | 340.85 | 1.41 | 4.43 | 9.58 | 0.01 | 23 |
| Lithuania | 278.55 | 189.22 | 1077.28 | 0.72 | 314.10 | 1.07 | 3.18 | 4.42 | 0.11 | 23 |
| Latvia | 224.27 | 108.79 | 1262.90 | 0.08 | 287.95 | 2.08 | 8.27 | 43.24 | 0.00 | 23 |

Source: the database provided by the "Our world in data" organisation.

Similar to evaluating the changes in the averages of RCA indices, RCA's relationship to COVID-19 cases in a particular Baltic country is also being studied and compared in trade with the external and EU domestic markets.

## 4. Results

The averages of our estimates and RCA indices by industries are presented in Table 4. The Baltic states do not appear to have a comparative advantage in trade with third countries (their RCA values in the Balassa index were found to be lower than 100).

**Table 4.** RCA averages and t-test statistics of Baltic States' exports to the EU and outside the EU.

| Industry | Year | Estonia Inside EU Average | t-test | Estonia Outside EU Average | t-test | Latvia Inside EU Average | t-test | Latvia Outside EU Average | t-test | Lithuania Inside EU Average | t-test | Lithuania Outside EU Average | t-test |
|---|---|---|---|---|---|---|---|---|---|---|---|---|---|
| 1 | 2017 | 154.8367 | | 40.65416 | | 126.3612 | | 68.78282 | | 78.46699 | | 30.93657 | |
| | 2018 | 103.6514 | 5.39543 * | 29.46134 | 3.540633 * | 97.47715 | 2.972708 * | 48.47894 | 3.00027 * | 94.29869 | −2.131537 ** | 24.31831 | 2.048306 *** |
| | 2019 | 90.55873 | 7.375604 * | 31.6263 | 2.347161 ** | 96.54105 | 2.807551 ** | 57.47854 | 1.726591 *** | 52.9432 | 5.854792 * | 21.63387 | 3.770687 * |
| | 2020 | 99.16226 * | 6.071004 * | 22.71382 | 4.886903 * | 113.0523 | 1.328641 | 67.78984 | −0.001157 | 67.97722 | 2.301828 ** | 27.86583 | 1.312287 |
| | 2021 | 86.91937 | 7.615107 * | 21.38554 | 6.27 * | 265.2249 | −2.48186 ** | 48.28696 | 2.946549 * | 69.87545 | 1.607064 | 15.43924 | 6.344968 * |
| | 2017 | 15.66742 | | 0.743439 | | 11.38587 | | 10.12023 | | 17.7086 | | 53.45741 | |
| | 2018 | 9.965516 | 3.829238 * | 0.560585 | 2.276415 ** | 14.13226 | −0.85341 | 11.50434 | −0.761821 | 17.96049 | −0.220474 | 63.55714 | −1.069101 |
| | 2019 | 6.756572 | 7.875949 * | 0.374685 | 4.108555 * | 3.986599 | 10.88441 * | 13.6115 | −1.162045 | 12.43356 | 4.025287 * | 67.65712 | −1.16489 |
| | 2020 | 11.15602 | 3.10502 * | 0.41566 | 3.916291 * | 5.734732 | 7.673981 * | 8.386579 | 0.876766 | 12.7207 | 5.099221 * | 68.44194 | −0.628153 |
| | 2021 | 5.941241 | 8.205501 * | 0.01046 | 10.36285 * | 10.92875 | 0.213643 | 6.893347 | 1.473021 | 13.74125 | 2.919533 * | 32.30755 | 2.360195 ** |
| 3 | 2017 | 342.7376 | | 67.02971 | | 148.4864 | | 40.77843 | | 126.7015 | | 15.37973 | |
| | 2018 | 237.8499 | 8.60345 * | 26.30254 | 10.61518 * | 109.5915 | 4.409328 * | 27.13522 | 3.105208 * | 113.7805 | 2.044099 *** | 16.92034 | −1.979674 *** |
| | 2019 | 215.1301 | 10.17135 * | 42.16715 | 6.087167 | 126.3978 | 2.307821 ** | 63.20329 | −1.850195 *** | 80.63209 | 8.140063 | 15.31005 | 0.07516 |
| | 2020 | 243.2371 | 7.63455 * | 33.13026 | 7.838926 * | 143.8715 | 0.532471 | 68.76855 | −2.649491 ** | 108.5505 | 2.962232 * | 18.12219 | −2.478955 ** |
| | 2021 | 175.3395 | 12.79748 * | 0.384747 | 18.01458 * | 263.3659 | −3.602188 * | 2.72973 | 8.970664 * | 91.33201 | 5.451644 * | 0.265006 | 39.2058 * |
| | 2017 | 162.3039 | | 20.74199 | | 73.45839 | | 15.01057 | | 77.04343 | | 22.07606 | |
| | 2018 | 145.8691 | 0.985658 | 8.654795 | 2.623096 ** | 76.31469 | −0.683637 | 23.59262 | −3.176402 * | 107.4234 | −4.830882 * | 31.11604 | −3.295656 * |
| | 2019 | 107.758 | 5.467633 * | 7.459182 | 3.936683 * | 84.69871 | −1.918112 *** | 19.29952 | −1.644837 | 62.35476 | 2.848337 * | 15.99352 | 2.781939 ** |
| | 2020 | 117.9911 | 4.153913 * | 3.873036 | 4.985956 * | 87.81156 | −2.758072 ** | 15.14165 | −0.050549 | 83.60375 | −0.896195 | 22.18491 | −0.034473 |
| | 2021 | 108.2399 | 4.811774 * | 0 | 6.161893 * | 122.4283 | −5.056093 * | 0 | 7.038681 * | 59.49967 | 4.190056 * | 0.877559 | 11.31652 * |
| 5 | 2017 | 153.2812 | | 2.018128 | | 46.68781 | | 1.255941 | | 46.68781 | | 1.255941 | |
| | 2018 | 138.2581 | 1.285383 | 4.103563 | −2.375389 ** | 59.2063 | −3.499886 * | 2.36501 | −3.424706 * | 59.2063 | −3.499886 * | 2.36501 | −3.424706 * |
| | 2019 | 127.2039 | 2.270861 ** | 3.697983 | −3.816343 * | 41.43496 | 1.799878 *** | 1.64967 | −1.371045 | 41.43496 | 1.799878 *** | 1.64967 | 1.371045 |
| | 2020 | 174.7691 | −1.30217 | 8.115351 | −0.892082 | 54.14568 | −2.212992 ** | 1.971183 | −2.397622 ** | 54.14568 | −2.212992 ** | 1.971183 | −2.397622 ** |
| | 2021 | 106.668 | 4.190951 * | 0 | 15.4195 * | 44.01619 | 0.977233 | 0.021327 | 4.881501 * | 44.01619 | 0.977233 | 0.021327 | 4.881501 * |
| | 2017 | 219.2899 | | 29.3707 | | 111.8671 | | 16.66813 | | 136.6746 | | 13.09718 | |
| | 2018 | 180.7565 | 3.753854 * | 34.9308 | −1.096775 | 85.81877 | 2.938698 * | 10.51194 | 2.710346 ** | 82.0631 | 3.053392 * | 12.99573 | 0.071861 |
| | 2019 | 151.4706 | 6.447491 * | 26.96352 | 1.189097 | 102.4332 | 0.952476 | 11.94451 | 1.749627 *** | 56.63916 | 4.587989 * | 11.14861 | 1.640429 |
| | 2020 | 166.9525 | 4.859963 * | 30.6261 | −0.431461 | 123.0373 | −1.293159 | 20.39962 | −1.146976 | 70.94111 | 3.76369 * | 11.32384 | 1.583813 |
| | 2021 | 123.2899 | 8.976863 * | 0 | 19.95635 * | 173.4864 | −4.035963 * | 0.00317 | 7.75004 * | 63.01103 | 4.195267 * | 0.376204 | 13.98919 * |
| 7 | 2017 | 185.6579 | | 33.61417 | | 119.2632 | | 32.41452 | | 80.62303 | | 16.88649 | |
| | 2018 | 134.5214 | 4.431296 * | 24.41144 | 0.476709 | 99.97703 | 2.376724 ** | 28.38648 | 0.786897 | 87.45436 | −1.248871 | 32.0989 | −4.567286 * |
| | 2019 | 92.68229 | 11.4019 * | 44.80678 | −0.568775 | 91.93024 | 3.430504 * | 35.65195 | −0.507309 | 59.79903 | 5.18573 * | 14.48711 | 1.231818 |
| | 2020 | 219.1295 | −1.626528 | 22.46416 | 0.584454 | 160.1149 | −1.97294 *** | 52.56559 | −2.416597 ** | 90.39575 | −2.135948 | 24.72109 | −2.655838 ** |
| | 2021 | 119.7298 | 6.586064 * | 0 | 1.766947 *** | 210.5865 | −3.946391 * | 5.00129 | 4.247017 * | 88.60712 | −1.174123 | 2.024061 | 5.994629 * |
| | 2017 | 5.579683 | | 0.029155 | | 11.96261 | | 0.004556 | | 3.797327 | | 0.043458 | |
| | 2018 | 12.09114 | −4.558403 * | 0.022432 | 1.950122 *** | 17.82279 | −1.524111 | 0.000231 | 1.269859 | 14.30602 | −4.205782 * | 0.114078 | −0.686014 |
| | 2019 | 11.38994 | −5.05558 * | 0.015982 | 5.474363 * | 18.65613 | −1.985851 *** | 0.003086 | 0.320079 | 8.77963 | −2.203767 ** | 0.008826 | 1.150572 |
| | 2020 | 9.094146 | −4.054391 * | 0.018372 | 4.567398 * | 7.645499 | 1.799374 *** | 0.002892 | 0.373009 | 10.54799 | −4.194157 * | 0.002672 | 1.366203 |
| | 2021 | 34.55856 | −1.471451 | 0 | 15.5325 * | 16.99324 | −1.681757 | 0 | 1.33897 | 8.99294 | −3.175371 * | 0 | 1.453601 |
| 9 | 2017 | 34.80432 | | 12.19498 | | 21.41696 | | 5.253568 | | 1.74967 | | 3.994861 | |
| | 2018 | 58.79169 | −2.681672 ** | 12.82948 | −0.177455 | 1.999554 | 3.373943 * | 4.091437 | 1.638473 | 1.573448 | 0.307629 | 5.971393 | −2.406064 ** |
| | 2019 | 138.472 | −5.796037 * | 15.02343 | −1.096865 | 3.430881 | 3.124956 * | 4.287402 | 1.230245 | 2.57645 | −1.244678 | 3.595033 | 0.75148 |
| | 2020 | 205.3747 | −10.41985 * | 13.64485 | −0.533061 | 2.400348 | 3.305097 * | 4.842962 | 0.501469 | 2.023207 | −0.378069 | 3.706777 | 0.569796 |
| | 2021 | 82.83002 | −2.319675 ** | 14.82979 | −1.008057 | 2.52527 | 3.28252 * | 3.737491 | 2.185964 ** | 2.322462 | −0.911982 | 4.000793 | −0.010337 |

* Denotes that the variables are significant at the 1% level; ** Denotes that the variables are significant at the 5% level; *** Denotes that the variables are significant at the 10% level.
Source: calculated by the authors.

We identified the highest RCA index values in Estonia in agriculture (1) and construction, housing, electrical appliance, and furniture (3) industries (40.65 and 67.03). In contrast, the same industries reached 68.78 and 40.78, respectively, in Latvia. On the contrary, Lithuania experienced lower RCA index values: 30.94 for agriculture (1) and 53.46 for mining, fuels, chemicals, electricity, and waste (2) industries.

It should be noted that the RCA index is calculated in aggregate for the entire sector, but this does not mean that the goods of the whole industry do not have a revealed comparative advantage. Those goods that have a high RCA index value are likely "hidden" under the goods that do not have it. A more in-depth analysis of the four-sign nomenclature would be required for greater precision, but this is beyond the scope of this study.

However, in trade with EU countries, the Baltic states had RCA index values higher than 100 in some industries. For Estonia, these were: agriculture (1); construction, housing, electrical appliances, and furniture (3); textile (4); transport (5); ICT and computers (6); health and pharmaceuticals (7); and other industries (9). Latvia reached above 100 in these industries, except transport (5) and other industries (9). Lithuania experienced above 100 RCA index value in three industries only: construction, housing, electrical appliance, and furniture (3); textiles (4); and ICT and computers (6).

The dynamics of RCA index values of Baltic countries suggest that recent values (that, in 2017, were higher than 100) had begun to decline a few years before the outbreak. This was the case for all mentioned industries of all three Baltic countries except for textiles (4) in Lithuania, which reached 107.42 in 2018 but rapidly declined. This may be related to a decrease in aggregate demand in 2019 (Gajdosikova et al. 2022). However, despite this decline in RCA index values, in the face of the pandemic, Baltic countries secured their competitiveness in most industries where RCA index values were very high immediately before the crisis. For example, in Estonia, only the agriculture (1) industry lost comparative advantage (RCA < 100). On the other hand, Lithuania lost this loss in ICT and computers (6) and construction, housing, electrical appliances, and furniture (3).

As mentioned, another part of our research focuses on the possible effect of COVID-19 cases on the competitive advantage of different industries in the Baltics. As can be seen from the regression analysis results (see Table 5), the COVID-19 pandemic had a clear difference in terms of its impact on RCA in trade with the EU and third countries. It is also clear that the Baltic states experienced different effects of the pandemic in various industries. The given coefficients reveal how much one additional case determines the change in the RCA index.

**Table 5.** The results of RCA and smoothed COVID-19 cases regression analysis.

| Industry | Baltics | | Estonia | | Lithuania | | Latvia | |
|---|---|---|---|---|---|---|---|---|
| | Inside EU | Outside EU | Inside EU | Outside EU | Inside EU | Outside EU | Inside EU | Outside EU |
| 1 | | −0.016136 ** | | | | −0.009036 *** | 0.294312 * | −0.023752 ** |
| 2 | | | −0.005742 ** | −0.000377 * | | | 0.012503 * | |
| 3 | | −0.02855 * | −0.071474 ** | −0.031322 * | | | 0.214963 * | |
| 4 | | | | −0.003558 * | | | 0.052999 ** | −0.010602 *** |
| 5 | | | −0.079672 * | | | | 0.084280 * | −0.002208 ** |
| 6 | | −0.014871 * | −0.056845 * | −0.027925 * | | | 0.085385 * | |
| 7 | | −0.018605 ** | −0.086368 *** | −0.020212 * | | | 0.168758 * | |
| 8 | | −0.0000746 ** | | −0.0000173 * | | | | |
| 9 | | | | | | | | −0.001982 *** |

* Denotes that the variables are significant at the 1% level; ** Denotes that the variables are significant at the 5% level; *** Denotes that the variables are significant at the 10% level. Source: calculated by the authors.

By analysing each country individually, the pandemic's impact on the competitiveness of each of the Baltic states in trade with EU countries is significantly higher than in trade with third countries. Moreover, one can see that the pandemic has had the most widespread impact on Estonia: RCA index value was negatively affected in industries such as mining, fuels, chemicals, electricity, and waste (2); construction, housing, electrical appliances, and furniture (3); ICT and computers (6); and health and pharmaceuticals (7). Meanwhile, the RCA index value in textile (4) and government and military (8) industries negatively

impacted trade with third countries only, whereas this only affected trade with EU countries in transport equipment, travel, and postal services industries (5).

Meanwhile, Lithuania and Latvia have proven to be more resilient to the effects of the pandemic. As such, Lithuania has not experienced the apparent impact of the pandemic in terms of RCA index values except for the agriculture industry (1). Meanwhile, Latvia has experienced growth regarding RCA index value in trade with EU countries. Moreover, in Latvia's case, the pandemic's negative impact has only been manifested in agriculture (1), textile (4), and transport industries (5).

By analysing Baltic countries in general, it should be noted that the OLS regression analysis provides statistically significant estimates only for trade with third countries. Therefore, there are no statistically significant estimates of how COVID-19 cases might impact the EU. However, in our cautious view, this does not necessarily mean that the results may be different when examining the impact of COVID-19 cases on intra-EU trade in the Baltic states. On the contrary, this may indicate that the Baltic states' trade within the EU, where they are more competitive (the RCA often exceeds 100), has been less impactful due to the pandemic-caused crisis. Therefore, we suggest that, as a result, there are no statistically significant COVID-19 cases and RCE in terms of the total intra-EU trade data of the Baltic states.

In summary, the whole picture suggests that highly competitive industries in the Baltic states demonstrated resilience against the pandemic by the end of 2021. However, temporary declines were observed in some industries (e.g., agriculture (1) in Latvia). In 2021, Estonia and Latvia secured competitiveness (RCA > 100) in five industries, whereas Lithuania lost it in all industries. This could be explained by Lithuania having not experienced high competitiveness before the pandemic crisis started.

## 5. Discussion

The Baltic states have experienced significant transformation in the manufacturing sector over the past three decades. The systemic change has been analysed extensively in line with economic growth (Hübner 2011), integration with the European Union (EU), the competitive advantage of high-technology goods export (Falkowski 2018), and the effects of the global crisis of 2008 (Kattel and Raudla 2013). The transformation of the manufacturing sector impacted the Baltics' economic development by contributing to output, employment, and exports. Studies confirm that the manufacturing industry for many economies remains the critical engine of economic growth through higher demand for capital and investment and utilisation of domestic human capital (Su and Yao 2016). It has also been a significant component in the transformation of smaller economies, which, in fact, highly rely on the demand in the international market (Sabonienė 2011). At the same time, industries of small open economies are more exposed to fast-growing international competition and more vulnerable to changes in the global market. According to the study performed by Nehrebecka (2021), almost all industries will experience the impact of COVID-19, sections being hit particularly hard will involve services that, due to the ban on gatherings of people and the recommendation to avoid crowds, will lose most of their revenue and will fail to make up for this loss in the future.

Cieślik et al. (2016) noted that the manufacturing industry of the CEECs (including Baltic states) became highly integrated into the global value chains (GVC). Cieślik et al. (2019) confirmed the positive impact of the economic potential of conditions affecting foreign companies' business opportunities on the role of CEECs in international trade relations and more stable and deep connections within GVCs. Even though Baltic countries are neighbouring countries, they have significant differences, particularly in trade patterns and comparative advantages, mainly due to their different manufacturing bases and differences in administrative reforms and policy frameworks that were implemented (Bernatonytė and Normantienė 2009). Further studies on the competitiveness of the Baltic states confirm that Lithuania, Latvia, and Estonia not only compete with one another but also specialise in manufacturing quite similar exporting goods. For example, the competitive profile of

Latvia and Lithuania was quite identical in textiles and clothing, and Estonia's and Latvia's similarities were recorded in the raw materials industry. It has also been observed that all three Baltic states during the period of 1999–2012 specialised in more labour-intensive and material-intensive goods, as well as resource-intensive goods with little research-intensive or capital-intensive goods (the results of RCA within 1999–2012 were negative or relatively insignificant) (Misztal 2009; Pilinkienė 2014; Remeikienė et al. 2015; Landesmann et al. 2015; Laaser et al. 2015; Andrews and Serres 2016). That is why it has been highlighted that all three Baltic states developing a substantial competitive advantage in international trade need to focus their exports on more high-value-added knowledge-intensive, high-skill-labour intensive and technology-intensive goods (Sabonienė 2011). This statement, proven in the Nehrebecka study (2021), that the probability of default over a 12-month horizon after the start of COVID-19 pandemic was higher for companies with a low level of technology (e.g., trading) than for medium- and high-tech companies. The pandemic generated a technological transformation in service delivery, with the latest technological developments having profound implications for national productivity, posing technology shocks that have formed a major challenge to underdeveloped economies (Saif et al. 2021). Lithuanian authors also underline the importance of the smart specialization strategy (Dzemydaitė 2021).

Some scholars suggest that the globalisation process, apart from its benefits to global trade, has also played a significant negative role in supply chain vulnerabilities and, in fact, increased the impact of potential disruptions (Christopher et al. 2011; Mena et al. 2022). Globalisation during the pandemic revealed its downside for countries and companies. As countries became interdependent, they became more exposed to systemic supply chain risks (Scheibe and Blackhurst 2017; Mena et al. 2022). Thus, COVID-19 reshaped the general pattern of globalisation (Mena et al. 2022). In line with the development of globalisation and competition growth between countries, it became more and more difficult for smaller countries to maintain and gain a competitive advantage. Differences between countries are becoming more pronounced (Nowak et al. 2020; Boikova et al. 2021).

Hagemejer and Mućk (2018) analysed the growth potential of domestic value-added. They pointed out that while technological backwardness is felt in CEE countries, capital stocks, technology imports and FDI inflows have increased in eastern European countries, which focus on modern, export-oriented production. Nevertheless, Boikova et al. (2021) distinguished differences between Baltic States.

The practice of developed countries indicates that advanced technologies, innovation, and R&D are critical industries for the economic development of an economy and the improvement of international competitiveness (Sabonienė 2011; Nehrebecka 2021; Saif et al. 2021). For vulnerable economies (Mizik et al. 2020; Ullah et al. 2021), it is recommended to focus more on self-efficiency, internationalisation, regional financial integration, increasing the level of product processing, domestic exports, and regional development by concentrating on the export of higher value-added products and by mainstreaming the export of products with competitive potentials, strategic communication between partners in the region, pragmatic cooperation and a friendly exchange approach worldwide, coordination and cooperation in foreign and local affairs and shared prosperity in the particular region and globally.

## 6. Conclusions

Economic activity—especially international trade in global conditions—requires a new strategic approach and fundamental transformation to help ensure the resistance of different sectors to external shocks. One such shock was the COVID-19 pandemic, which disrupted the established world order. The spread of COVID-19 has revealed that, due to the rapidly growing interdependence of flows of goods and services, countries have become particularly vulnerable in an economic sense, and governments must be able to reorganise their international trade strategies to reduce the vulnerability to such global economic shocks.

Our study revealed that the COVID-19 pandemic's impact on each Baltic country's competitiveness in trade with EU countries is significantly higher than in trade with third countries. The Baltic states did not have a comparative advantage in trade with third countries. Although Lithuania and Latvia proved to be more resilient to the consequences of the pandemic, industries with lower levels of RCA index values were more affected.

Meanwhile, in trade with EU countries, many Baltic industries had an RCA index value of over 100 in 2017. It should be noted that these same values began to decline a few years before the pandemic's start, coinciding with a decline in aggregate demand (Gajdosikova et al. 2022). In our view, this may have been a reflection of trends of economic crises following Brexit and other reasons, which were accelerated by the onset of the pandemic. However, clear conclusions in this regard require additional research.

It is important to note that the Baltic states (especially Estonia and Latvia) secured their competitiveness in many industries, as evidenced by their high RCA index values immediately before the pandemic. In other words, the highly competitive industries of the Baltic states showed remarkable resilience to the impact of the pandemic. However, a short-term decrease in the RCA index values was observed in individual cases.

Despite this paper's contribution to the existing literature, the following limitations must be highlighted. First, the scope of the article includes only the Baltic states. It would be worth extending the research and examining other EU countries (or groups of EU countries) in more detail to assess the whole picture with additional analysis of individual product groups. Furthermore, future research could examine extended periods, allowing more post-pandemic changes to be observed. Finally, in addition to the RCA index calculation, the gravity model could also be applied to assess the positions of the countries' trading partners in the international market and the demand they influence following the COVID-19 outbreak.

On the other hand, the changes caused by the COVID-19 pandemic serve as an excellent opportunity to test (and possibly propose new) economic theories and reveal the peculiarities of individual countries and industrial sectors in international trade and FDI attraction. Coquidé et al. (2022) emphasized that the impact of COVID-19 clearly showed that the industrial production in USA and EU countries should be increased in order to become more self-sufficient and independent—a fact that should be taken into consideration by policy makers. Li et al. (2023) investigated that logistics among other factors had the most severe impact on sales during pandemics. Opening up the logistics of sales channels is the primary policy choice. Moreover, storage warehouses and insurance are also important pre-emptive measures (Li et al. 2023). As the world moves faster and faster into the era of the digital economy, the specialisation map of countries, industrial sectors, and companies after the COVID-19 pandemic will inevitably differ from what it previously was. Therefore, identifying significant causes, factors, and altered processes can supplement and (or) change our knowledge of critical international trade economic phenomena.

**Author Contributions:** Conceptualisation, J.D., A.B., and V.C.; methodology, A.B.; validation, A.B., J.D., and V.C.; formal analysis, A.B.; investigation, J.D. and V.C.; resources, J.D., A.B., and V.C.; data curation, A.B.; writing—original draft preparation, J.D., A.B., and V.C.; writing—review and editing, J.D.; visualisation, A.B.; supervision, J.D.; project administration, V.C.; funding acquisition, V.C. All authors have read and agreed to the published version of the manuscript.

**Funding:** This research received funding from the Research Council of Lithuania (LMTLT), agreement No S-MIP-21-28.

**Informed Consent Statement:** Not applicable.

**Data Availability Statement:** The data presented in this research paper are available upon request from the corresponding authors.

**Conflicts of Interest:** The authors declare no conflict of interest.

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
