# Peer review of "The Impact of the COVID-19 Pandemic on the Revealed Comparative Advantage of Industries in the Baltic States"

_economies, doi:10.3390/economies11020047_

Round 1
Reviewer 1 Report
The authors conduct an impressive amount of empirical work, I feel that the contribution is not sufficient to merit publication due to lack of new results and insights given the large amount of literature in COVID-19.
Author Response
Thank you for your comment. We have supplemented the literature list and, accordingly, the discussion part with sources that are most related to the topic under consideration.
Reviewer 2 Report
General remarks:
1. Paper is quite short which does not make it to be more clear. Please look at:
- "COVID-19: stress-testing non-financial companies: a macroprudential perspective. The experience of Poland", Eurasian Economic Review 11 (2), 283-319
2. Motivation for additional factor is not clear.
3. Methodological section is very long and still it may be misleading.
It is not worth describing the ANOVA analysis, it is a basic statistical method and known.
Don't confuse regression analysis with ANOVA analysis.
It is worth actually conducting a regression analysis to have an added value of a given article.
Author Response
- Thank you for your comment. We have supplemented the literature list and, accordingly, the discussion part with sources that are most related to the topic under consideration.
- Motivation for additional factor is not clear. Response: we have expanded the theoretical part of the article.
- Methodological section is very long and still it may be misleading. Response: we shortened the methodological part and better structured it into two separate stages of the study. We transferred the entire literature analysis that was in the methodological part to the theoretical part of the article. ANOVA now is not being mentioned. We separated entirely both analysis we employed in the research.
Reviewer 3 Report
This manuscript addresses an interesting and timely issue, is generally well written and it thus can be considered for publication in Economies.
However, I have a few comments which need to be elaborated before acceptance:
1. From where in table 2 in the column Obs. values ​​above 2000, since this is based on monthly data from 2017-2021?
2. What is the reason for such a large, outlier value of Max - 16877 and kurtosis - 2248 for Latvia outside the EU27?
3. Analyzing the results from Table 4, it can be seen that more significant values ​​determining the impact of Covid on the RCA values ​​are in the outside EU columns. So why the statement: "The pandemic's impact on the competitiveness of each of the Baltic states in trade with EU countries is significantly higher than in trade with third countries."? The same also appears in the abstract: "This study revealed that the COVID-19 pandemic's impact on each Baltic country's competitiveness in trade with EU countries is significantly higher than in trade with third countries."
The table 4 seems to be showing the opposite of what is written? Please clarify.
Author Response
Thank you very much for your valuable comments, the article has been adjusted to take them into account:
- From where in table 2 in the column Obs. values ​​above 2000, since this is based on monthly data from 2017-2021? Response: good notice – that was a mistake. Corrected.
- What is the reason for such a large, outlier value of Max - 16877 and kurtosis - 2248 for Latvia outside the EU27? Response: Thank you for the notice – mistake was corrected (see table 2).
- Analyzing the results from Table 4, it can be seen that more significant values ​​determining the impact of Covid on the RCA values ​​are in the outside EU columns. So why the statement: "The pandemic's impact on the competitiveness of each of the Baltic states in trade with EU countries is significantly higher than in trade with third countries."? The same also appears in the abstract: "This study revealed that the COVID-19 pandemic's impact on each Baltic country's competitiveness in trade with EU countries is significantly higher than in trade with third countries."The table 4 seems to be showing the opposite of what is written? Please clarify.
Response: In our cautious view, this does not necessarily mean that when examining the impact of COVID-19 cases on intra-EU trade of the Baltic states, the results may be different. This may indicate the fact that the Baltic states' trade within the EU, where they are more competitive (the RCA often exceeds 100), has been less impactful by the pandemic caused crisis. As a result, there is no statistically significant COVID-19 cases and RCE in terms of the total intra-EU trade data of the Baltic states.
Round 2
Reviewer 1 Report
The authors' revised manuscript is much improved, and I would suggest that the authors cite the following references as they are very relevant to the research question and could increase the paper's research impacts:
1. Bao, Zhengyang, and Difang Huang. "Shadow banking in a crisis: Evidence from FinTech during COVID-19." Journal of Financial and Quantitative Analysis 56, no. 7 (2021): 2320-2355.
2. Baqaee, David, and Emmanuel Farhi. "Keynesian production networks and the covid-19 crisis: A simple benchmark." In AEA Papers and Proceedings, vol. 111, pp. 272-76. 2021.
3. Li, Nan, Muzi Chen, and Difang Huang. "How Do Logistics Disruptions Affect Rural Households? Evidence from COVID-19 in China." Sustainability 15, no. 1 (2022): 465.
4. Wouters, Olivier J., Kenneth C. Shadlen, Maximilian Salcher-Konrad, Andrew J. Pollard, Heidi J. Larson, Yot Teerawattananon, and Mark Jit. "Challenges in ensuring global access to COVID-19 vaccines: production, affordability, allocation, and deployment." The Lancet 397, no. 10278 (2021): 1023-1034.
Author Response
Dear reviewer,
thank you for the suggested articles. We included them in the literature analysis.
Best wishes
Author
Reviewer 2 Report
Dear Author,
I accept.
Author Response
Thank you